# Revisiting the evolution of downhill thunderstorms over Beijing: A new perspective from radar wind profiler mesonet

Xiaoran Guo[a,b], Jianping Guo[a*], Tianmeng Chen[a], Ning Li[a], Fan Zhang[a], Yuping Sun[a],

[a]*State Key Laboratory of Severe Weather, Chinese Academy of Meteorological Sciences, Beijing 100081, China*

[b]*College of Earth and Planetary Sciences, University of Chinese Academy of Sciences, Beijing 100049, China*

Correspondence to:

Dr./Prof. Jianping Guo (Email: jpguo@cma.gov.cn; jpguocams@gmail.com)

**Abstract**

Downhill thunderstorms frequently occur in Beijing during the rainy seasons, leading to substantial precipitation. The accurate intensity prediction of these events remains a challenge, partly attributed to insufficient observational studies that unveil the thermodynamic and dynamic structures along the vertical direction. This study provides a comprehensive methodology for identifying both enhanced and dissipated downhill thunderstorms. In addition, a radar wind profiler (RWP) mesonet has been built in Beijing to characterize the pre-storm environment downstream to the thunderstorms at the mountain foot. This involves deriving vertical distributions of high-resolution horizontal divergence and vertical motion from the horizontal wind profiles measured by the RWP mesonet. A case study of enhanced downhill thunderstorm on 28 September 2018 is carried out for comparison with a dissipated downhill thunderstorm on 23 June 2018, supporting the notion that a deep convergence layer detected by the RWP mesonet, combined with the enhanced southerly flow, could favor the intensification of thunderstorms. Statistical analysis based on radar reflectivity from April to September 2018–2021 have shown that a total of 63 thunderstorm events tend to be enhanced when entering the plain, accounting for about 66% of the total number of downhill thunderstorm events. A critical region for intensified thunderstorms lies on the downslope side of the mountains west to Beijing. The evolution of the downhill storm is associated with the dynamic conditions over the plain compared to its initial morphology. The existence of strong westerly winds and divergence in the middle of troposphere exert a critical influence on the enhancement of convection, while low-level divergence more leads to the dissipation. The findings underscore the significant role of RWP network in elucidating the evolution of downhill storm.

## Short Summary

The prediction of downhill thunderstorm (DS) remains elusive. Here we propose an objective method to identify DS, based on which enhance and dissipated DS are discriminated. A radar wind profiler (RWP) mesonet is used to derive divergence and vertical velocity. The mid-troposphere divergence and prevailing westerlies enhance the intensity of DS, whereas the low-level divergence is observed when the DS dissipates. The findings highlight the key role that RWP mesonet plays in the evolution of DS.

## 1. Introduction

The complex evolution of convective systems crossing mountainous terrain represents a substantial forecasting challenge. It has been previously reported that downhill thunderstorms with intensive reflectivity and good organization are more likely to successfully maintain or strengthen compared to isolated and small-scale thunderstorms (Castro *et al.*, 1992). Various thermal factors that favor the development of downhill thunderstorm have been identified, including higher instability and lower convective inhibition (Letkewicz and Parker, 2010, 2011; Keighton *et al.*, 2007), adequate water vapor accompanied by low-level jets (Tompkins, 2001; McCaul and Cohen, 2004; Weckwerth *et al.*, 2014), and cool pool (Teng *et al.*, 2000; Jeevanjee and Romps, 2015; Li *et al.*, 2017; Xiao *et al.*, 2017). Furthermore, a few studies in the literature have demonstrated the importance of the dynamic environment over the plain, such as surface and low-tropospheric convergence for convection initiation (Frame and Markowski, 2006; Miglietta and Rotunno, 2009; Wilson *et al.*, 2010), and strong vertical wind shear (Parker *et al.*, 2007; Reeves and Lin, 2007; Xiao *et al.*, 2019).

The topography in Beijing is intricate, given its location at the foot of the Taihang Mountains to the west and the Yan Mountains to the north, both of which have ridges with elevations exceeding 1200 meters (Figure 1a). Wilson *et al.* (2007) found that downhill thunderstorms, particularly those originating from the west, constituted 79% of all thunderstorms in Beijing between 2003 and 2005, as determined through a statistical analysis of thunderstorm datasets. The distinctive topography and the frequent occurrence of downhill thunderstorm in Beijing afford us an excellent opportunity to observe the inherent dynamic structures of downhill thunderstorms and their pre-storm environments. This, in turn, allows for a more in-depth investigation into the potential physical mechanisms underlying the formation of this severe weather event. However, most of the previous studies are limited to the analysis of a single downhill thunderstorm case (Chen *et al.*, 2017; Sun and Cheng, 2017; Kang *et al.*, 2019). Besides, the investigation of pre-storm environment and evolution process of thunderstorm are either based on the model simulation (Chen *et al.*, 2005; Xiao *et al.*,

2015; Li *et al.*, 2017) or reanalysis data (Wang *et al.*, 2019), largely owing to the dearth
of high-density continuous vertical profiling measurements of wind, temperature, and
humidity.
Furthermore, there exist no objective method that can be used to identify and track
the propagation of downhill thunderstorm in the literature Therefore, more urgent
efforts are warranted to investigate the difficult-to-forecast storm type from a statistical
perspective of ground-based atmospheric profiling mesonet observations.  A high-
density mesonet, consisting of six radar wind profilers (RWP) has been established in
the Beijing since 2018 (Figure 1b) to continuously observe three-dimensional wind
fields with high temporal and vertical resolution. This provides us with a valuable tool
to explore the atmospheric dynamic structures, such as areal averaged vorticity,
divergence, and vertical velocity, of the pre-storm environment for the downhill
thunderstorms by using the parameters derived from the RWP mesonet. The primary
goals of this study are twofold: (1) to develop an objective method to identify the event
of downhill thunderstorm and its evolution, mainly based on composite radar
reflectivity from weather radar; and (2) to explore the statistical patterns of downhill
thunderstorms and reveal the dynamical structures in the development of downhill
thunderstorms, aiming to attain a deeper understanding of the evolution processes of
these thunderstorms.
The next section describes the data and methodology, in which a novel objective
method is proposed to characterize the evolution of downhill thunderstorm. Section 3
presents a case study of an enhanced downhill thunderstorm. Statistical analyses of the
relationship is conducted in section 4 between wind profile, convergence and the
evolution of downhill thunderstorms. A summary and concluding remarks are given in
section 5.

## 2. Methodology and data

### 2.1. Identification of downhill thunderstorms

To study the downhill thunderstorms in Beijing, areas in Figure 2a is selected as the region of interest (ROI). Then, ROI is divided into three subregions by terrain height: Area to the west and north of the ridge line is defined as the mountainous region (ROI$_m$), marking as dark gray in Fig. 2a; and the area with surface elevation less than 100 m is defined as the plain region (ROI$_p$), marked with white; the light-gray area between these two lines is defined as the downslope region (ROI$_d$).

The flow chart for identifying downhill thunderstorms from composite radar reflectivity is illustrated in Figure 2b, which is mainly comprised of the following steps: Firstly, based on the well-established findings in literature from previous studies (e.g., Kingsmill, 1995; Weckwerth, 2000; Qin and Chen, 2017; Bai *et al.*, 2019), echoes with radar reflectivity reaching over 35 dBZ triggered in ROI$_m$ are identify as potential downhill thunderstorms. To eliminate false signals, those echoes with area less than 50 km$^2$ are filtered out.

Secondly, these potential clusters are tracked using the area overlapping method (Machado *et al.*, 1998; Huang *et al.*, 2018; Chen *et al.*, 2019). Noted that during merging processes, only the largest cluster is tracked continuously, while others are subsequently terminated. Likewise, during the splitting processes, only the largest cluster is tracked continuously, while others are attributed to newly initiated storms. Suppose the i$^{th}$ (i = 1, 2, ...) thunderstorm (i.e., $\boldsymbol{S_i}$) is observed in ROI$_m$ at time n (i.e., $T^n$), the properties of $S_i^n$ including the centroid ($C_i^n$), area ($A_i^n$) and maximum reflectivity ($MR_i^n$) are obtained.

Thirdly, the downhill thunderstorms are defined by whether the potential clusters move into ROI$_d$ and ROI$_p$. And if the centroid of $S_i$ crosses the ridge line and moves from ROI$_m$ to ROI$_d$ at time j, $T^j$ is defined as the starting time when $S_i$ begins to go down the hills. Similarly, if the centroid of $S_i$ crossed the plain line and moves from

ROI$_d$ to ROI$_p$ at time k, $T^k$ is defined as the arrival time when $S_i$ reaches the plain. Then,
$T^k$-$T^j$ is defined as the downhill duration of $S_i$. An example of $S_i$ is depicted in Fig. 2a.

Finally, the downhill thunderstorms are classified into two categories, the

enhanced downhill storms (EDS) and dissipated downhill storms (DDS). These two
subsets are classified by comparing the area and maximum reflectivity at the time $T^k$
to those at time $T^j$. If at least one of the criterions $A_i^k \cong A_i^j$ and $MR_i^k \cong MR_i^j$ fulfils, $S_i$
is considered as an EDS, otherwise it is defined as a DDS.

Most of previous research, either case studies or small sample statistics analysis,

lack an objective criterion used to determine downhill thunderstorms. They typically
focus on EDS in the presence of high-impact weather and less consider DDS. Compared
to the existing approaches in the literature, our methodology can discriminate between
these two types of downhill thunderstorms for its capability in defining the timing and
location of storms and tracking their corresponding evolution. Therefore, this
methodology can be readily applied to other regions with similar topography as long as
weather radar measurements are available.
*2.2. Meteorological data*

As depicted in section 2.1, radar reflectivity derived from the Doppler radar

network dataset with a grid resolution of 0.01° at 10-min intervals during the rainy
seasons (i.e., April−September) in 2018-2022 is used to identify downhill
thunderstorms over Beijing.

Upper-air sounding balloons launched at the Zhangjiakou (ZJK) and Beijing

Weather Observatory (BWO) (see their locations in Fig. 1b) are used to provide the
vertical thermodynamic features during the downhill thunderstorms. Generally, the
balloons launches twice a day at 0800, and 2000 Local Standard Time (LST), providing
the vertical profiles of temperature, pressure, relative humidity, and horizontal winds
with a vertical resolution of 5–8 m (Guo *et al.,* 2020). For the sake of improving the
prediction skill of summertime storm, an additional radiosonde launch is performed at
1400 LST daily at the BWO for the period from June 1 to August 31.

Ground-based meteorological variables, including 2-m air temperature ($T_{2m}$), dew

point temperature, and pressure measured at 5-min intervals and precipitation measured

at 1-min intervals from automated surface stations (AWSs) are also used in the analysis

over the study area.

Geopotential height at 500 hPa and horizontal wind at 850 hPa from the fifth

generation ECMWF reanalysis (ERA5) datasets derived by European Centre for

Medium-range Weather Forecasts (ECMWF) are used for analysing the large-scale

conditions in a case study of a heavy precipitation event in Beijing. The dataset has 37

pressure levels, which is made publicly accessible on a grid spacing of 0.25° at hourly

intervals (Hoffmann *et al.,* 2019).

*2.3. Radar wind profiler measurements*

The RWP mesonet in Beijing, as presented in Table 1 and Fig. 1b, consists of six

RWPs positioned at Shangdianzi (SDZ), Huairou (HR), Yanqing (YQ), Haidian (HD),

Pinggu (PG), and BWO. The RWPs used in this study are CFL-6 Tropospheric Wind

Profilers, manufactured by the 23rd Institute of China Aerospace Science and Industry

Corporation. These instruments provide sampling height, horizontal wind direction and

speed, vertical wind speed, horizontal credibility, vertical credibility, and refractive

index structure parameter. And the data are recorded at 6-min intervals at 34 levels with

a vertical resolution of 120 m below 4 km above the ground level (AGL) in low-

operating mode, and at 25 levels with a vertical resolution of 240 m from 4 to 10 km

AGL in high-operating mode (Liu *et al.*, 2019). Considering that the six RWPs located

at different terrain heights, the horizontal velocities measured by each RWP are

interpolated to the same altitude, starting from 0.5 km above mean sea level (AMSL)

with a vertical resolution of 120 m.

Dynamic parameters, such as the horizontal divergence profiles can readily be

calculated by vertical wind profile measurements derived from soundings or RWPs

distributed along the perimeter of a circle or a triangle over an area (Bellamy, 1949;

Carlson and Forbes, 1989; Lee *et al.*, 1995; Bony *et al.*, 2019). The reliability of the

measurements and triangle method is demonstrated in the previous work (Guo *et al.*,

2023). Thus, we also employes this methodology to calculate the regional mean divergence, vorticity and vertical velocity profiles within the triangular regions built by the RWPs mesonet.

**3. A case study of an EDS event**

EDSs present significant challenges for local weather forecasters in accurately predicting the intensity of precipitation during nowcasting. In this section, an observational case study of this type of downhill thunderstorm is selected to explore the role of thermodynamic and dynamic environment on the evolution of the downhill thunderstorms.

This storm originated from the $ROI_m$ and began to go down the hill at 1200 LST of 28 September 2018, then hit Beijing after approximately 2–3 hours. Several AWSs in the Yanqing District recorded lightning activity and hails accompanied with an hourly rainfall amount of over 30 mm from 1430 to 1530 LST. It is noteworthy that the intensity of downhill thunderstorm became weakened before 1400 LST but intensified as it approached the plain area of Beijing.

*3.1. Synoptic background*

Sounding taken at the ZJK (Figure 3a) at 0800 LST located in the westerly flow sector, showed a surface-based temperature inversion below 900 hPa and a deep dry layer aloft from 850 hPa up to about 400 hPa. At the same time, similar temperature and humidity stratification was seen at the BWO (Figure 3b) with little convective available potential energy (CAPE) of 170.8 J kg$^{-1}$ and convective inhibition (CIN) of 61 J kg$^{-1}$. The veering of a northwesterly wind to a westerly wind from 850 hPa to above 600 hPa indicated the presence of cold advection at 0800 LST. Unfortunately, no sounding was available to elucidate the temporal evolution of atmospheric thermodynamic and dynamic environments during the passage of EDS from 1200 LST to 1500 LST. We can only speculate that the thermal stratification seems insufficient to facilitate the initiation and subsequent organization of deep convection, even though

considering the possible enhancement of unstable layer as the mixed layer grew after
0800 LST.
Then, we resort to the synoptic pattern from ERA-5 reanalysis at hourly intervals.
At 500 hPa (Figure 3c), the large-scale conditions at 1400 LST on 28 Sep 2018 was
characterized with a deep cold vortex at the border of Mongolia and China, and Beijing
was situated in the cold sector, with a cold center approximately 500 km to the south,
and influenced by strong westerly flows. At 850 hPa (Figure 3d), a trough extended
from northeast to southwest over $ROI_d$, resulting in significant southwesterly flow prior
to the trough over Beijing. The veering of a southwesterly wind at 850 hPa to a westerly
wind 500 hPa indicated the presence of warm advection. The changeover from cold
advection at 0800 LST to warm advection at 1400 LST in the lower troposphere could
account for the subsequent deepening organization of convection after the thunderstorm
entered the plain.
*3.2. Radar reflectivity and surface observations*
Radar reflectivity at 1200 LST (Figure 4a) showed that a convective line with
several convective cores was detected across the ridge line and moved gradually
southeastward into $ROI_d$ driven by the low-level northwesterly flows. Surface
streamlines evidently showed dominant west-to-southwesterly surface winds in $ROI_m$
and south-to-southwesterly flows in $ROI_p$ (also see Figure 4a). In downslope regions,
the local mountain-valley orientations appeared to account for up-valley flows in
various directions. A surface analysis at 1200 LST, given in Figure 5a, shows a humid
center in the northwest of the mountain region due to the previous precipitation,
whereas the relative humidity of the downslope and plain was less than 60%. The
thermal boundary near the ridge line which generated by the terrain could also be seen.
$T_{2m}$ over the plain area was on average of greater than 20 °C, whereas the mean $T_{2m}$
over the mountainous region was less than 10°C. The large northwest-southeast-
oriented temperature gradient appeared to account for the intensification and better
organization of the at 1230 LST (Figure 5b). Surface convergence emerged ahead of
the convective line, indicated by the streamlines in Figure 4b, which were associated
with a pre-squall mesotrough/mesolow.
At 1300 LST, convective line with reflectivity exceeding 35 dBZ had spitted into
two segments (Figure 4c). The northern segment was completely separated from the
main storm in the southwest and then expanded northeastward by the intersecting
streamlines, with another convective cell initiated near the local converging center
around 117°E, 41.5°N before 1330 LST (Figure 4d). The southern segment maintained
with the total rainfall exceeding 10 mm from 1300 to 1400 LST. Meanwhile, the wet
center gradually moved eastward to the northeast of the mountain region (Figure 5c-d).
Until 1400 LST, the convective cells started to merge into a linear convective system,
and the frontal edge of the convection line had arrived at triangle 1 with weaker
intensity than before (Figure 4e).
Further, we attempt to examine the roles of cold pool and low-level wind shear in
maintaining the intense squall line in accordance with the theory of Rotunno et al.
(1988). However, it's difficult to perform a comprehensive and quantitative analysis
due to the inhomogeneous environment and measurement. Here, we qualitatively use
the horizontal winds over YQ (Figure 6a) to estimate vertical wind shear (VWS) om
the downslope and $T_{2m}$ to identify a cold pool (Figure 5). At 1300 LST, the wind speed
below 1.5 km AMSL was weaker than 5 m s$^{-1}$ while was stronger than 15 m s$^{-1}$ above
2.5 km AMSL. The maximum value of VWS occurred at the altitude of 1.8 km AMSL
with the value exceeding 20 m s$^{-1}$ km$^{-1}$. In less than 10 minutes, cold downdrafts
produced a sharp drop in $T_{2m}$ by 6°C in the south of the convective cells (Figure 5c-d).
The effects of the resulting low-level VWS might balance with those of the cool pool,
which helped stimulate the development of more intense storms from 1300 to 1330 LST.
Meanwhile, the accompanying evaporative cooling in the descending flows
strengthened the cold pool. After 1330 LST, horizontal wind speeds in the lowest 2 km
layer strengthened to shrink the low-level VWS to about 10 m s$^{-1}$ km$^{-1}$. The cold-pool-
induced horizontal vorticity could overpower that of the low-level wind shear, partly
facilitating the dissipated radar echo before 1400 LST (Figure 5e). Moreover, this might
be related to the relatively strong cold pool located in the south, which potentially cut
off the warm southerly inflow from the plains to the mountains. Then, cool pool
weakened with convection and the overpowering effect diminished.

As the storm approached $ROI_p$ from 1400 LST, composite radar reflectivity shows

that it was significantly strengthened to an intense and well-organized squall line
(Figure 4e-4g). AWSs within triangle 1 captured its associated rainfall. Abrupt increase
in surface pressure by +3 hPa was seen across the gust front in the triangle 1 when the
maximum rainfall rate exceeded 3mm $(6min)^{-1}$ (not shown). Except for the above-
mentioned balanced state between cool pool and low-level vertical wind shear, this
enhancement could potentially be associated with the dynamic lifting over plain area .
Due to the disadvantage of surface observations in monitoring the vertical dynamic
features, we have to resort to the examination of the evolution of high-resolution
divergence and vertical velocity derived from the fine-scale RWP mesonet in the
following subsection.
*3.3. Divergence and vertical velocity*

Before the convective system reached the plain area, sustained southwesterly wind

above 2 km AMSL was observed after 1200 LST at YQ (Figure 6a), which was likely
driven by the synoptic pattern, accompanied with upper-layer divergence and
downdraft in triangle 1 (Figure 6b). The much weaker near-surface southerly wind and
unnoticeable divergence could to a certain extent be influenced by the valley flows at
the foot of the mountains. Meanwhile, a peak of positive vorticity exceeding $10^{-4}$ $s^{-1}$
and a deep layer of negative vorticity up to 5 km AMSL in triangle 1 were maintained
during this time period (Figure 6c). Then, pronounced southerly wind occurred after
1300 LST that corresponded to the rapidly intensification in convergence below 2 km,
providing an uplifting background, albeit less than 0.1 m $s^{-1}$. This updraft assisted the
upward transport of moist air in the planetary boundary layer (PBL), which facilitated
the subsequent formation of clouds and convective rainfall. Additionally, a vorticity
maximum near $3 \times 10^{-4}$ $s^{-1}$ at 1348 LST in the PBL might also be favorable for
organized convective development.
The low-level wind speeds over YQ started to increase to 10 m s$^{-1}$ as a result of
the downward momentum transport. The subsequent enhancement in convergence
coincided well with the intensification of southwesterly winds (>10 m·s$^{-1}$) up to 3 km
ASML after 1418 LST. Such intensification in convergence and updraft were also well
captured by triangles 2 (not shown), even with more than one hour in advance of the
convective rainfall arrival. Upward motion in triangle 1 increased in amplitude and
deepened rapidly in depth as the squall line propagated southeastward, and triggered
rainfall over triangle 1. The most intense convergence occurred at 1430 LST and
extended from 1 km to above 2.5 km AMSL afterwards as a result of latent heat release
during cloud formation. The maximum vertical velocity reached 0.35 m s$^{-1}$ around 3.5
km AMSL, which were about 6 min prior to the peak area-averaged rainfall rate at 1448
LST. The significant convergence diminished after 1454 LST, when deep convection
moved out of triangle 1 (Figure 4h). Downdrafts are found with moderate upward and
downward motions in the stratiform area.
Interestingly, as the squall line propagated eastward and approached the urban
center after 1500 LST, it rapidly dissipated as the area of convective echo was reduced
by a scale fact of 4/5 until 1600 LST (not shown). This appeared to result from the
blocking of water supply by the high risings over the Beijing's built-up area, the so-
called "urban bifurcation" effects on moving thunderstorms (Changnon, 1981; Zhang,
2020). In this case, deep convection in the urban center and northern suburban area
were suppressed due to the urban blocking effects. It was consistent with the persistent
low-level divergence over triangle 3 and 4 with the maximum value of $3 \times 10^{-4}$ s$^{-1}$
occurring near surface (not shown). Clearly, this result can help understand the urban
building-barrier induced divergence and the dissipation of thunderstorm.

## 4. Comparison with a DDS event

In the preceding section, low-level convergence is an effective signal for the
maintenance of an EDS event. In this section, we present a DDS event that occurred on
23 June 2018 in attempt to investigate the difference of pre-storm environment for two
types of downhill thunderstorms. Similar to the trajectory of the EDS, the DDS began
to go downhill at 1600 LST (Figure 7a) and then propagated southeastward with the
area larger than 1000 km$^2$. It had dissipated rapidly upon reaching the plain of Beijing
after 1900 LST and diminished until 2100 LST.

Figure 8a shows the SkewT/Log P diagram derived from the sounding taken at the

BWO at 1400 BJT, 23 June 2018. It can be seen that a dry troposphere was presented
in the early afternoon. As the time lapsed, the humidity above 700 hPa increased at
2000 BJT (Figure 8b), even though the surface was characterized by a dry layer near
the surface. The surface relative humidity was less than 40% with $T_{2m}$ exceeding 30°C
and the dew point temperature less than 20°C. The CIN slightly decreased from 280.2
J kg$^{-1}$ at 1400 LST to 264.0 J kg$^{-1}$ at 2000 LST. By comparison, The CAPE increased
from 35.5 J kg$^{-1}$ at 1400 LST to 483.0 J kg$^{-1}$ at 2000 BJT.  As shown in Figure 8c, the
study area was situated to the west of the high-pressure ridge at 500 hPa and influenced
by northerly flows in front of the ridge, whereas the lower levels were dominated by
weak southwesterly winds below 850 hPa.

In the next, we examine the dissipation stage of downhill storm when it reached

triangle 1 with a focus on the evolution of atmospheric dynamic variables. A sustained
near-surface southeasterly winds was found over YQ before 1900 LST from the surface
streamlines and vertical wind profile that are shown in Figures 7b, c and 9a. The low-
level troposphere over triangle 1 was dominated by distinct deep divergence (Figure 9a)
and positive vorticity (Figure 9b) below 2 km AMSL. The deep divergence of regional
flows and larger CIN more tended to suppress the vertical motion breaking through the
resistance of a stable atmosphere (Xiao et al., 2019).

As the downhill thunderstorm reached YQ at 1900 LST (Figure 7d), the near-

surface wind turned into weak northwesterly winds accompanied by the rapid
intensification of convergence over triangle 1 under the force of the convective system
itself. The strongest convergence of this event with a value of $-3.8 \times 10^{-4}$ s$^{-1}$ below 1
km AMSL at 1906 LST. It is worth noting that the divergence layer above 1.5 km AMSL
persisted during the occurrence of precipitation after 1924 LST. Even though there were
the cyclone motion and weak updrafts with the maximum vertical velocity reaching 0.1
m s$^{-1}$, it was not enough to penetrate the divergence layer and lift the vapor to the lifting
condensation level (LCL) at around 800 hPa as shown in Figure 8b. The maximum
composite radar reflectivity of the echo sharply decreased from 64.5 dBZ at 1900 LST
to 53.5 dBZ at 2000 LST with the area shrinking by half (Figure 7e). The rainfall was
terminated which was consistent with the dominated low-level divergence until 2100
LST (Figure 7f).
The above comparison indicates that a linear system was intensified in to the squall
line with fast speed in front of a shortwave troughs in the EDS event. In the DDS event,
some scattered convective cells were organized into clusters as they propagated to the
plain under a weak ridge and then dissipated. For these two cases for EDS and DDS
event, the thermal stratification indicated the presence of unfavorable pre-storm
environmental settings with insufficient unstable energy and inadequate moisture. The
dynamic condition played a pivotal role for convective development during the passage
of the downhill thunderstorm. Compared with the DDS event, the enhanced southerly
winds and corresponding convergence in the lower level were distinct features of the
EDS. The above results indicate that the RWP mesonet could capture well the vertical
profiles of horizontal divergence and vertical motion, favorably supporting the
detection of convection.
Notably, small-scale variations of airflow in the narrow valley at the intersection
of Mt. Taihang and Mt. Yan undoubtedly impacts the dynamics of the EDS and DDS
event (Xiao et al., 2017). In other words, the storms from northwest need to pass by the
downslope, valley, and then upslope to reach the plain. The complex local terrain should
be taken into account the factors for the evolution of thunderstorms during the
southeastward propagation. However, the current resolution of observations is not
capable of resolving the dynamic processes associated with the convective development
in that region. We hope further explore this factor with the help of the numerical
simulation in the future.

## 5. Statistical results

### 5.1. General features of downhill thunderstorm events

To obtain a more robust understanding of the climatology for downhill thunderstorm evolution in Beijing, an in-depth statistical analysis is carried out in this study. According to the methodology mentioned in Section 2.1, we firstly identify a total number of 95 downhill thunderstorms triggered in $ROI_m$ and moved into $ROI_d$ and $ROI_p$ in the study area (Figure 1b) based on the radar reflectivity datasets during the rainy seasons (i.e., April- September) in 2018-2022. We perform a statistical analysis of the occurrence number of radar reflectivity that is equal to or greater than 35 dBZ on a grid spacing of 0.01° at 10-min intervals during these downhill thunderstorm events.

As shown in Figure 10a, downhill thunderstorms tend to initiate in $ROI_d$ with strong steep slopes near the ridges of the Yan Mountains associated with solar heating in the afternoon. The highest-frequency center is found mainly over the western downhill area extending to the plain with the occurrence number exceeding 400, due possibly to the large amount of eastward propagation of thunderstorms driven by the westerly or southwesterly flows during the warm seasons in Beijing (Chen *et al.*, 2012, 2014).

For all downhill thunderstorms, the relationship between the initial area and length-width ratio of thunderstorms at the beginning and the relative variation of area to the time it arrives at $ROI_p$ is analyzed. Here, we record the maximum (minimum) axis length of the radar echo with reflectivity ≥35 dBZ as the length (width) of the downhill thunderstorm, respectively. The area and length-width ratio tends to reflect the horizontal scale and organization of convective storms. Generally, linear convective storms show a length-width ratio greater than or equal to 3.0 (Chen and Chou, 1993; Meng *et al.*, 2013; Yang *et al.*, 2017). The results show that several mature thunderstorms with the area larger than 5000 km$^2$ tend to dissipate during the downhill process with weaker intensity and area, which are likely due to the splitting processes (Figure 10b). Convective lines commonly intensify to the squall lines, but several isolated and loose thunderstorms expand rapidly during the downhill process with

increasing area when entering the plain, which may be associated with the favorable
regional-scale lower tropospheric environment.
It is found that 63 thunderstorms events tend to be enhanced after it moved into
the downhill and urban areas, accounting for about 66 % of the total number of downhill
thunderstorms events, whereas 32 thunderstorm events tend to be dissipated. Most of
the DDSs arrive at the plain area in mornings and late afternoons (Figure 10c).
Specifically, 11 and 18 DDSs arrive at the plain area during the period of 0600–1200
and 1600–0000 LST which account for 34% and 56% of all DDSs, respectively. In
contrast, the EDSs tend to occur in early mornings and afternoons. 18 and 43 EDSs
arrive at the plain area before 0800 LST and after 1400 LST, respectively,
corresponding to the percentage of 26% and 68%.  Meso-scale circulations driven by
the urban heat island (UHI) effect and topography may contribute to the difference of
downhill storms' duration. As presented by Dou et al. (2015), the magnitude of UHI of
Beijing at the nighttime are stronger than in daytime. In the early morning, low-level
westerly and northwesterly winds converged into the Beijing's plain area because of a
combination of downslope mountain breezes and strong-UHI-induced convergence,
which accelerate the speed of thunderstorms towards the plain. The weaker
southeasterly upslope valley breezes in the late afternoon and evening make downhill
storms slow down and contribute to the prolonged duration. One caveat is that the
conclusions may vary by the number of available sample cases.
*5.2.  Dynamic conditions*
We present the trajectories and their moving directions of two types of downhill
storms (Figure 11) to confirm that the western part of $ROI_d$ is a key area for the
development of downhill thunderstorms. To better understand the similarities and
differences between EDS and DDS from the perspective of ambient atmospheric
environment, three-dimensional dynamic structures derived from RWP mesonet are
analyzed. Variables including wind speed, vertical wind shear, u-component and v-
component of wind, divergence and vorticity profiles are used to provide information
of dynamic structures before the downhill thunderstorms arrive. Thus, we select 68
downhill thunderstorms, including 50 EDSs and 18 DDSs, which pass through triangle
1 to the plain among all 95 samples and focus on these meso-scale parameters from YQ
station and triangle 1 in the following discussions.
The mean vertical wind profiles two hours prior to the arrival of the thunderstorms
are investigated. Horizontal wind speed, vertical wind shear, u-component and v-
component from the RWP in YQ, and divergence and vorticity over triangle 1 are
calculated (Figure 12). Results indicate that wind speed preceding EDSs and DDSs is
about 5 m s$^{-1}$ below 1.5 km (Figure 12a). Much stronger horizontal winds with the
maximum wind speed exceeding 15 m s$^{-1}$ are observed in the 1.5-5 km layer in advance
of the EDS events, The VWS below 5 km AMSL have no significant differences
between EDSs and DDSs before their arrival (Figure 12b). But the VWS preceding
EDS events is little bit stronger than that preceding DDS events, which could be likely
associated the critical influence that high vertical wind shear exerts on convection.
EDSs and DDSs mainly appears under the near-surface southeasterly and prevalent
southwesterly low-level flow near the foothills. The persistent supply of water vapor is
key for the successful propagation to the plains of downhill storms but doesn't
determine the enhancement or dissipation of convection. Notably, the average v-
component of wind decreases to near-zero above 3 km AMSL. The existence of stronger
westerly flow above 3 km AMSL is a favorable condition for the intensification of
downhill storms (Figure 12c), which well corroborates the results from case study.
The mean vertical structure of divergence and vorticity are given in Figure 12e and
f. Before the arrival of downhill storms, one can see the presence of weak divergence
near the surface due to the weak wind. Compared with EDSs, the divergence around
1.5-3 km AMSL is more evident near the arrival of DDSs with the maximum value of
$10^{-4}$ s$^{-1}$. When thunderstorms pass by, the strong divergence in the low level is not
conducive to the extension of upward movement within the boundary layer which
attributes to the dissipation of storms, especially when instability and moisture supply
are unfavorable. In contrast, the high-level divergence at around 4-5 km altitudes
promotes the compensation of the moist air and the upward transport heat, which

ultimately reinforce the storm. The vorticity field in Figure 8f is characterized by cyclonic flows at lower-levels and anticyclonic flows at midlevel, which is possibly dependent on the synoptic forcing. The vorticity prior to EDSs seems to be stronger than that of DDSs, the cooperation between lower-level cyclones and less divergence of convective system tends to promote the maintenance of updrafts, leading to heavy rainfall.

In the previous work, it has been confirmed that these dynamical variables derived from the RWP mesonet in Beijing provide strong supports for machine-learning-based prediction of severe convection (Wu *et al.*, 2023). The results therein show that the usage of RWP observational data as the random forest model input tends to result in better performance in the rainfall/non-rainfall forecast 30 min in advance of rainfall onset than using the ERA5 reanalysis data as inputs. In the future, these dynamic observations and methodologies need to be further incorporated into machine learning model for improving the prediction skill of downhill thunderstorms.

## 6. Summary and concluding remarks

Given the large uncertainty in prediction and huge impact, here we revisit the evolution of downhill thunderstorms and concurrent ambient atmospheric dynamic structures as derived from a high-density radar wind profiler (RWP) mesonet in Beijing. This RWP mesonet in Beijing is shown to be capable of continuously observing the horizontal wind fields in the lower troposphere with ultra-high vertical and temporal resolutions. It follows that the profiles of vertical wind shear, divergence and vorticity are derived from the triangle algorithm, which are used to analyze the pre-storm dynamic environment for the downhill storms.

First of all, a novel objective methodology has been developed to identify and track the downhill thunderstorms. Combined with the changes in area or intensity of radar echoes, enhanced downhill thunderstorms (EDSs) and dissipated downhill thunderstorms (DDSs) are discriminated. A case study of an EDS during the period of 1200-1500 LST of 28 September 2018 is performed. Of interest is that the intensity of

downhill thunderstorm became weaker before 1400 LST but intensified as it
approached the plain area of Beijing. Meanwhile, we present a DDS event that occurred
on 23 June 2018 in attempt to investigate the difference of pre-storm environment for
two types of downhill thunderstorms. For these two cases of EDS and DDS, the thermal
stratification indicated the presence of unfavorable pre-storm environmental settings
with insufficient unstable energy and inadequate moisture. The dynamic condition
played a pivotal role for convective development during the passage of the downhill
thunderstorm. Compared with the DDS event, the enhanced southerly winds and the
corresponding convergence in the lower level were distinct features of the EDS. The
above results indicate that the RWP mesonet could capture well the vertical profiles of
horizontal divergence and vertical motion, favorably supporting the detection of
convection.
To obtain a robust result concerning the evolution characteristics of the downhill
thunderstorms in Beijing, an in-depth statistical analysis is merited. The beginning and
arrival time of a downhill thunderstorm event are defined as the moment when the
centroid crosses the ridge line and plain, respectively. A total of 95 downhill
thunderstorms events occurring in the study area are identified based on the datasets of
radar reflectivity at 10-min intervals during the rainy season (i.e., April- September) of
2018–2022. The high occurrence frequency center of convection is found mainly
resides west to Beijing' plain area. And the area variation of convection is not sensitive
to the initial morphology itself. It is found that 63 thunderstorms tend to be enhanced
with larger area or radar reflectivity after it moved into the downhill and urban areas,
accounting for about 66 %. The statistical analysis indicates that most of the downhill
thunderstorms affect the plains in the morning and late afternoon. Most downhill
processes last about two hours while thunderstorms from the northwest and the north
may take a longer time possibly due to the further distance.
Thus, we illustrate the statistical analysis of dynamic quantities, such as horizontal
winds, vertical wind shears derived from the RWP at the mountain foot, and divergence
and vorticity derived from the west-most triangular region in the RWP mesonet, in

relation to the enhanced and dissipated downhill storms. Results indicate that much stronger westerly winds are observed in 1.5-5 km layer in advance of the EDS events and exert a critical influence on the development of storms. Furthermore, divergence at around 4-5 km altitudes promotes the compensation of the moist air and the upward transport heat, which ultimately reinforce the storm. Weaker lower-level divergence and cyclonic flows over the plain contribute to the development of robust updrafts and closer coupling between boundary layer and clouds, which favor the intensification of downhill thunderstorms.

Continuous measurements of the accurate dynamic quantities will make it possible to enable a more critical and quantitative evaluation for the development of downhill thunderstorms in the future. Nevertheless, the above-mentioned dynamic features, which are necessary to diagnose the evolution of thunderstorms, are not adequate to fully characterize the environment in which downhill storms are embedded. In particular, more explicit analysis of thermodynamic parameters, such as CAPE, K index, precipitable water, will be performed to characterize the pre-storm environments in detail.

**Data Availability**

We are grateful to ECMWF for providing ERA5 hourly data (https://www.ecmwf.int/en/forecasts/datasets/reanalysis-datasets/era5/). The radar wind profiler data are obtained from the National Meteorological Information Center of China Meteorological Administration (https://data.cma.cn), which can be only accessed via registration.

**Acknowledgments**

This work was supported by the National Natural Science Foundation of China under Grants of 42325501, U2142209 and 42105090. Last but not least, we appreciated tremendously the constructive comments and suggestions made by the anonymous reviewers that significantly improved the quality of our manuscript.

## Author Contributions

The study was completed with close cooperation between all authors. JG designed the research framework; XG and JG performed the analysis and drafted the original manuscript; JG, TC, NL, FZ and YS helped revise the manuscript.

## Completing interests

The authors declare that they have no conflict of interest.

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

**Table 1.** Summary of six radar wind profilers in Beijing.

| Station Name | Acronym | Lat. (°N) | Lon. (°E) | Alt. (m, AMSL) |
|---|---|---|---|---|
| Shangdianzi | SDZ | 40.66 | 117.11 | 286.5 |
| Huairou | HR | 40.36 | 116.63 | 75.6 |
| Yanqing | YQ | 40.45 | 115.97 | 489.4 |
| Haidian | HD | 39.98 | 116.28 | 46.9 |
| Pinggu | PG | 40.17 | 117.12 | 32.1 |
| Beijing Weather Observatory | BWO | 39.79 | 116.47 | 32.5 |



**Figures**

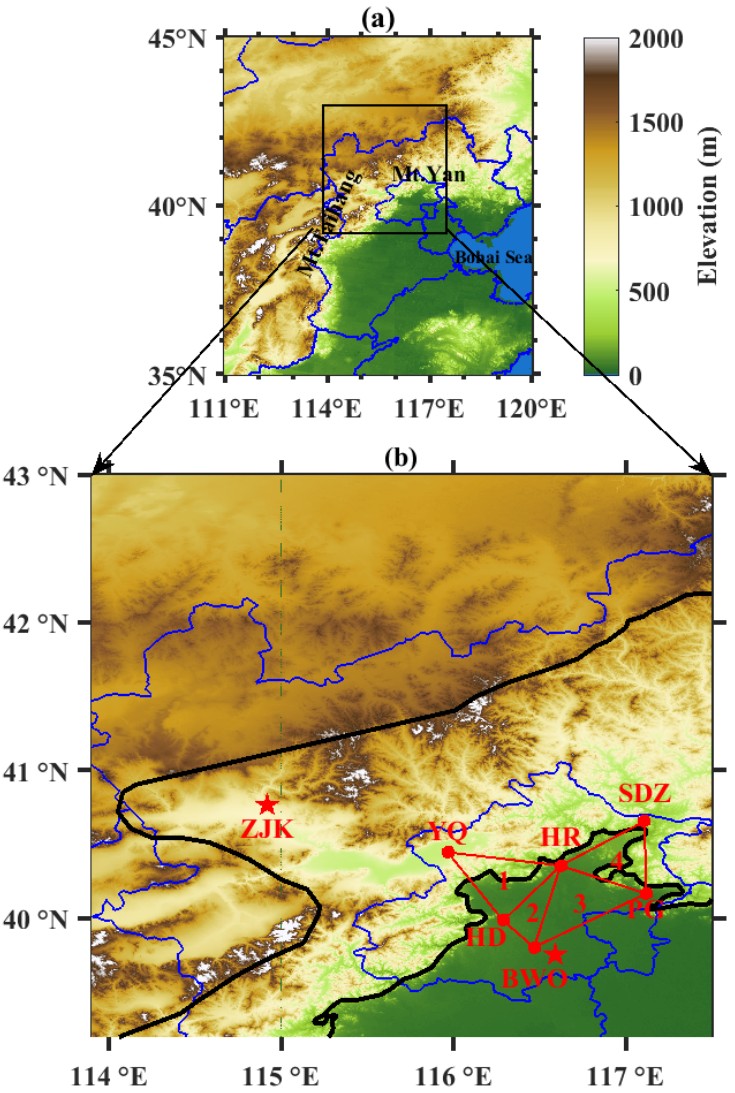

**Figure 1.** (a) Spatial distribution of the topography over northern China with the blue line denoting the Province boundaries. The locations of Taihang Mountains (Mt. Taihang), Yan Mountains (Mt. Yan) and Bohai Sea are written in black text. (b) Map of Beijing with six RWPs (red dots) deployed at Shangdianzi (SDZ), Huairou (HR), Yanqing (YQ), Haidian (HD), Pinggu (PG), and the Beijing Weather Observatory (BWO) and surrounding areas. The BWO and Zhangjiakou (ZJK) are deployed with an L-band radiosonde (red pentagrams). The four red triangles denote the areas used to calculate the horizontal divergence with the triangle method. The left black line mark the ridge line, and the right black line mark the plain line that denotes the 200-m terrain elevation.

764

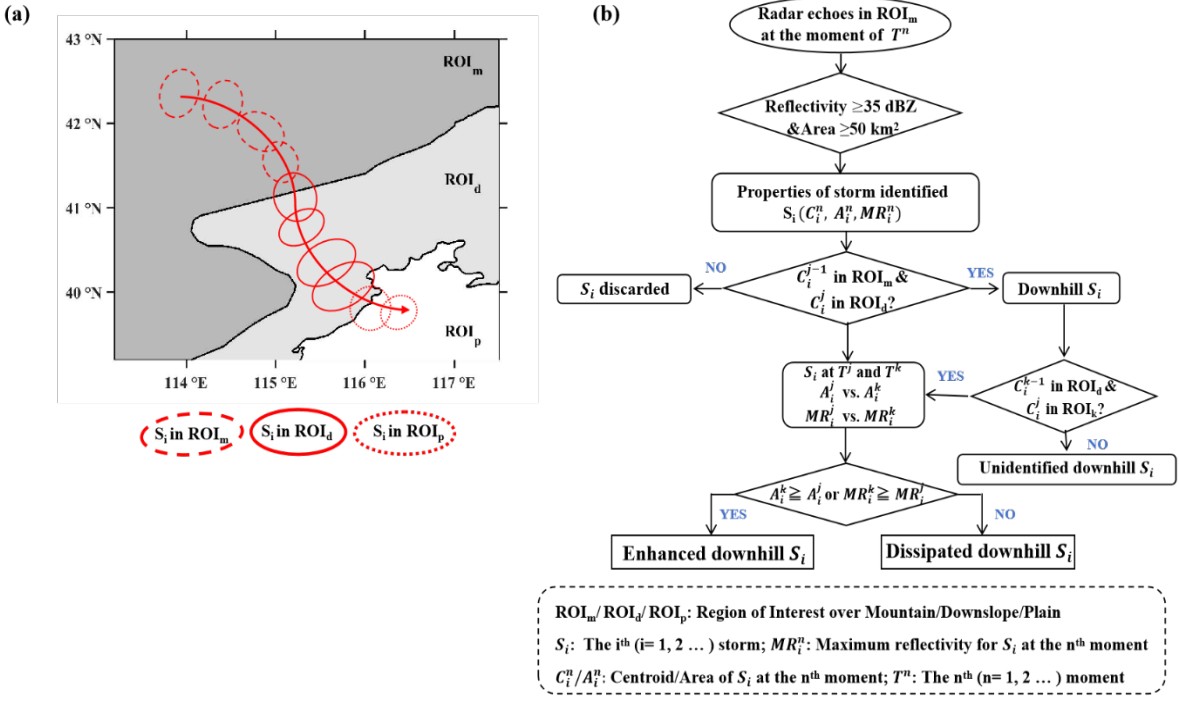

765

**Figure 2.** (a) Definition of the ROIs and the schematic diagram showing the track of a downhill thunderstorm $S_i$ (red circle). The red arrow denotes the trajectory of $S_i$. (b) Flow chart showing the primary processes to identify downhill thunderstorms in this study.

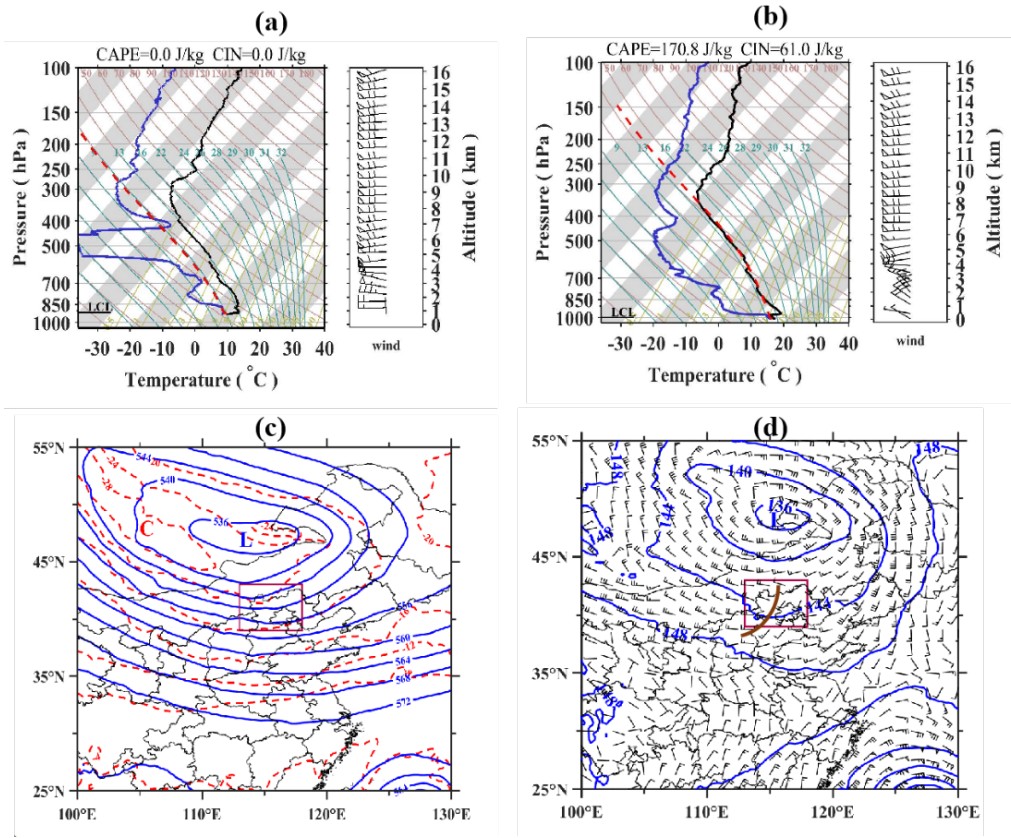

**Figure 3.** (a) SkewT/Log P diagram derived from the upper-air sounding at the ZJK at 0800 LST of 28 Sep 2018. (b) Same as (a) but for the upper-air sounding at the BWO. (c) Horizontal distribution of geopotential height at 500 hPa (solid blue lines at 40 gpm intervals) and temperature at 500 hPa (dashed red lines at intervals of 4 °C) at 1400 LST of 28 Sep 2018, both of which are obtained from the ERA5 hourly reanalysis data. The purple rectangle indicates the location of the study area shown in Figure 1b. Letters "L" and "C" denote the centers of a low-pressure system, and cold air, respectively. (d) Same as (c), but for the fields of geopotential height at 850 hPa (solid blue lines at 40 gpm intervals) and horizontal wind at 850 hPa. Note the distribution of a trough along the thick brown line.

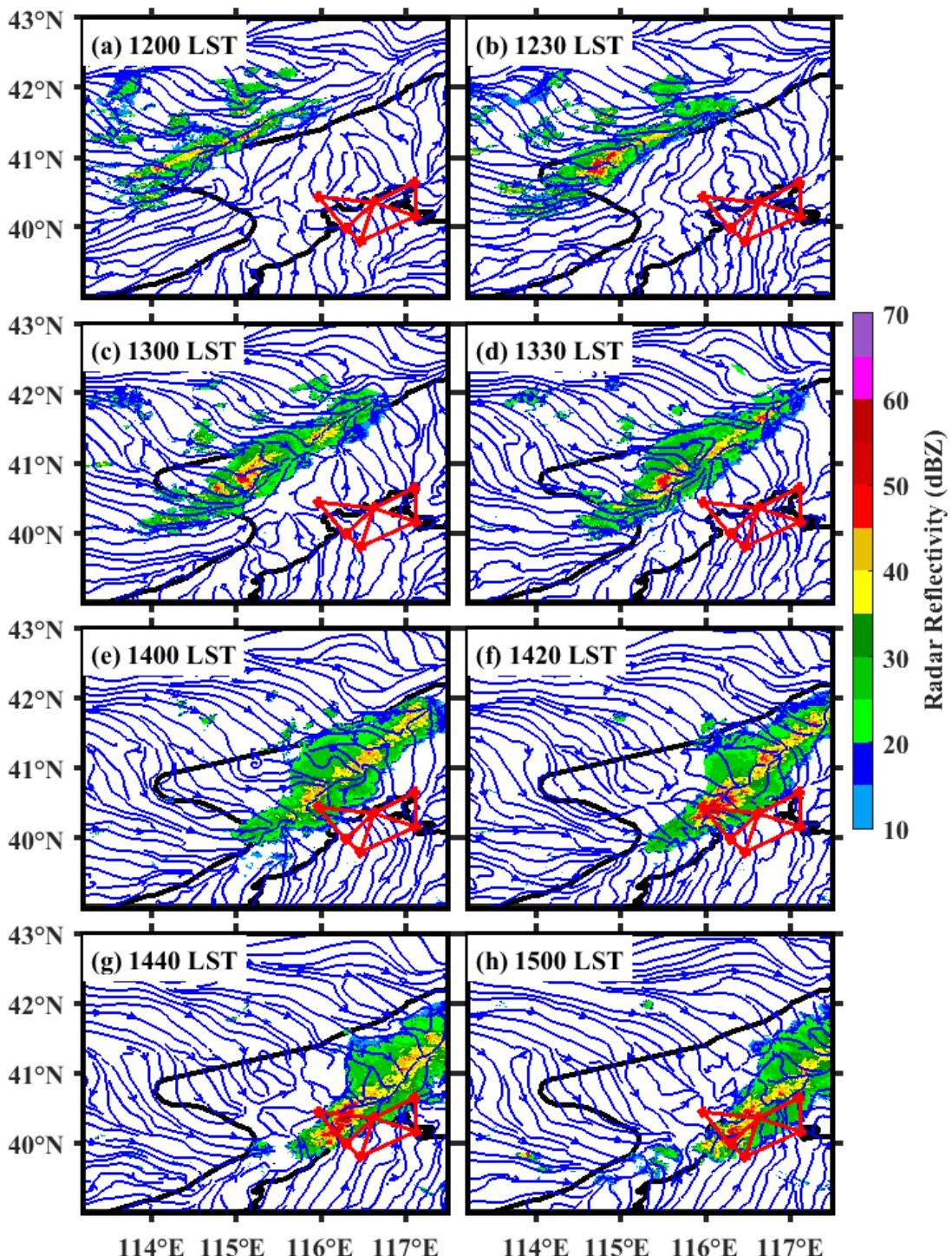

**Figure 4.** Evolution of the composite radar reflectivity (color-shaded, dBZ) and surface streamlines derived from AWSs for the case of an EDS event occurring during the period from (a) 1200 to (h) 1500 LST on 28 September 2018. The four red triangles denote the regions used to calculate the horizontal divergence and vertical motion with the triangle method. The two black lines mark the boundaries of the $ROI_m$, $ROI_d$ and $ROI_p$.

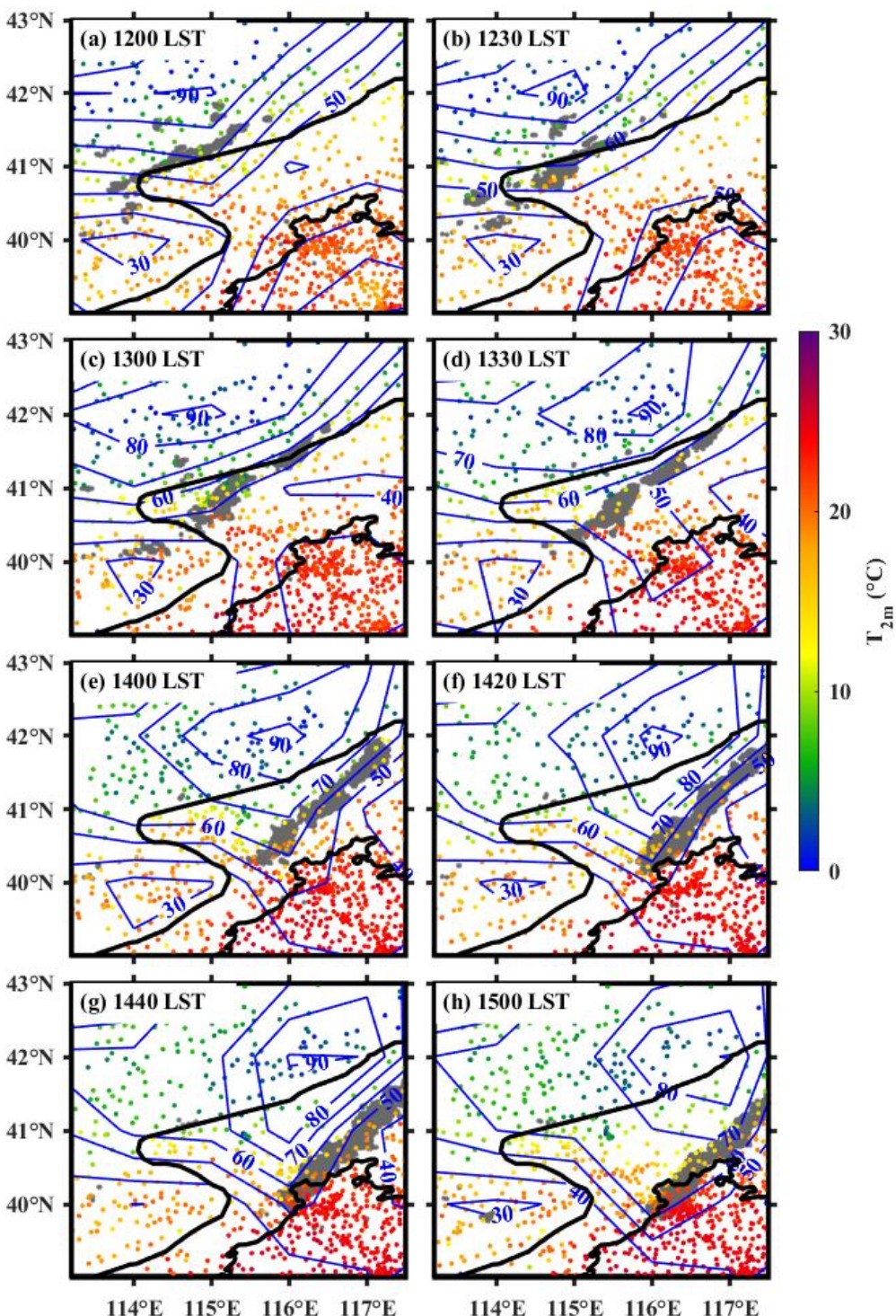

789

**Figure 5.** Evolution of the $T_{2m}$ (color-shaded, °C) and relative humidity (contour, %) derived from AWSs from (a) 1200 to (h) 1500 LST 28 Sep 2018. The left black line is the ridge line, the right black line is the plain line which denotes the 200-m terrain elevation. The gray region denotes the position of echo with radar reflectivity exceeding 35 dBZ.

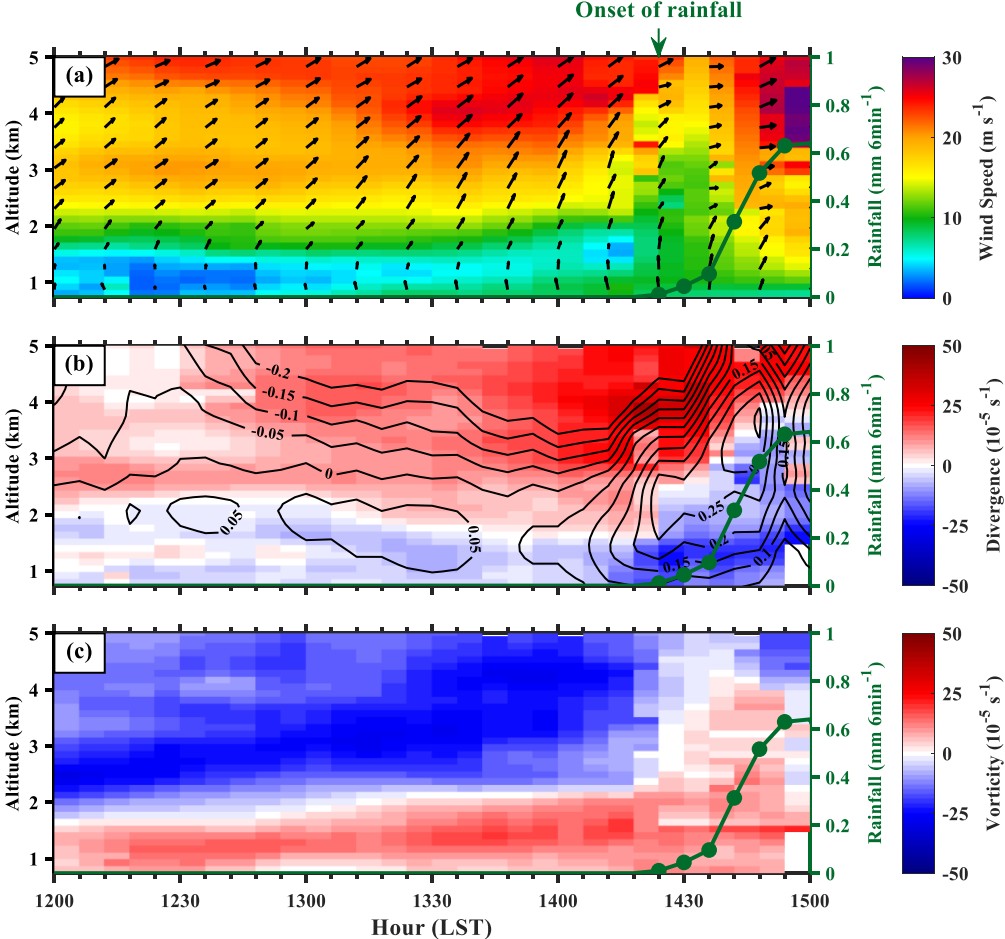

795

**Figure 6.** (a) Time series of horizontal wind vectors (m·s⁻¹) with wind speeds shaded

in the 0.5–5-km AMSL layer during the period of 1200–1500 LST 28 Sep 2018 at YQ

station. Green-dotted lines represent the triangle-area-averaged rainfall amount (mm

6min⁻¹) of triangle 1. (b) same as (a), but for the vertical profiles of the triangle-averaged

divergence (shaded, 10⁻⁵ s⁻¹) and vertical velocity (contour, m s⁻¹) for triangle 1. (c)

same as (b), but for the vertical profiles of the vorticity (shaded, 10⁻⁵ s⁻¹) for triangle 1.

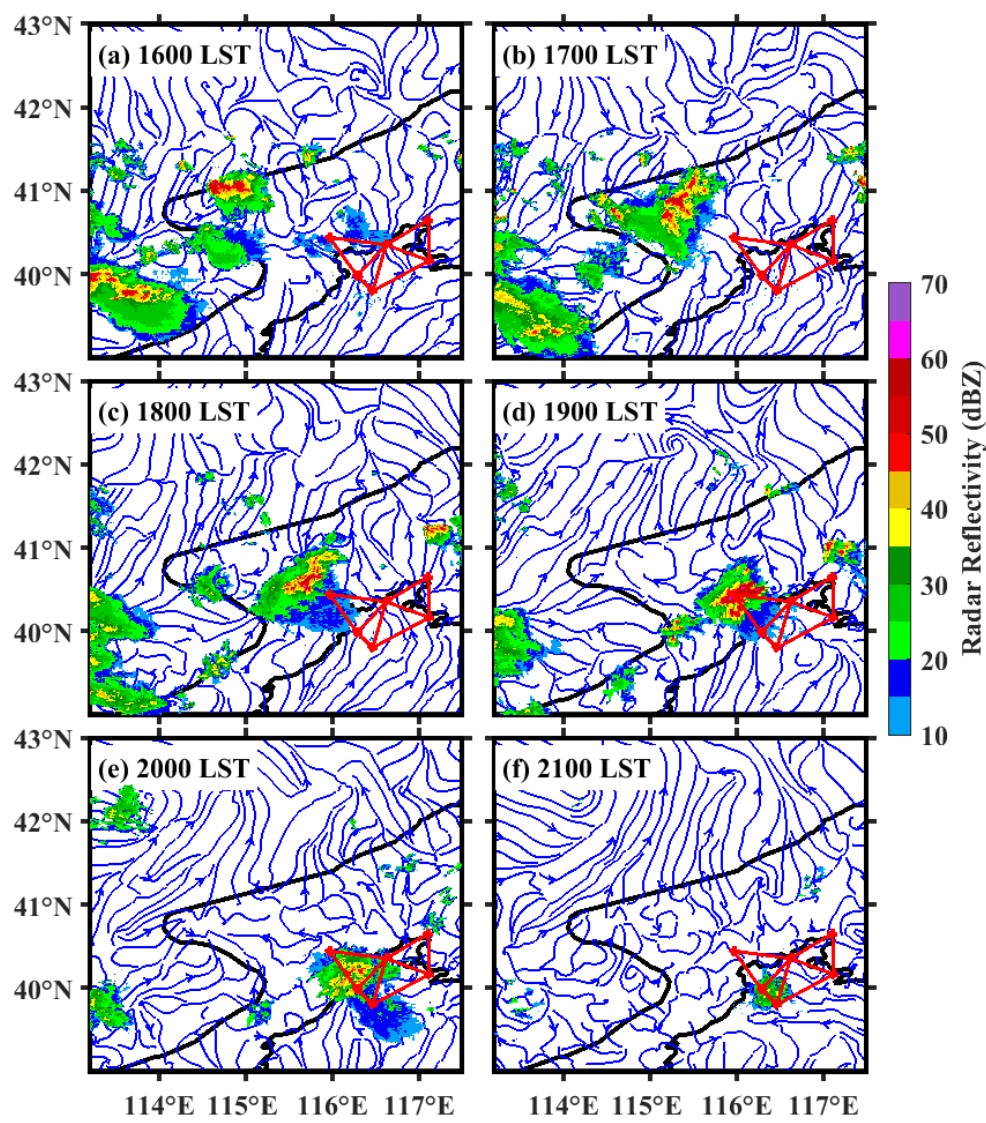


**Figure 7.** Same as Figure 4, but for the case of a DDS event occurring during the period

from (a) 1600 to (f) 2100 LST 23 June 2018.



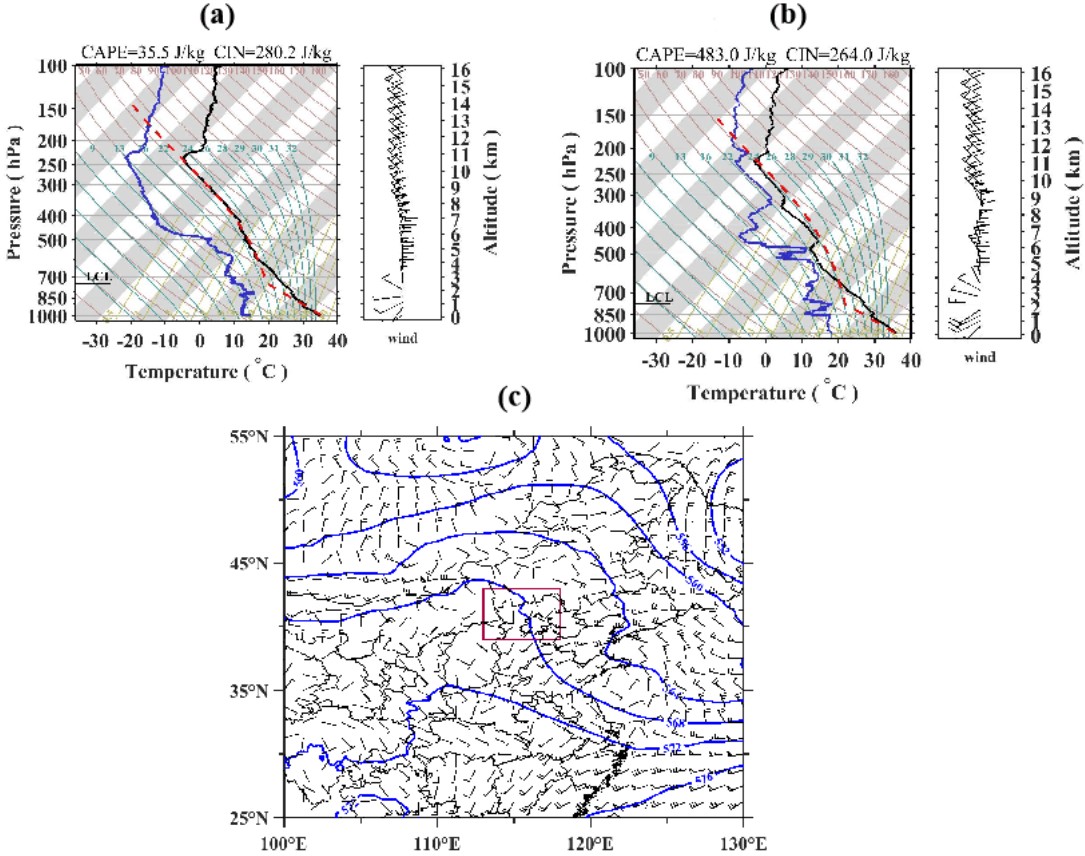


**Figure 8.** SkewT/Log P diagram derived from the upper-air sounding at the BWO at (a)
1400 LST and (b) 2000 LST of 23 June 2018. (c) Horizontal distribution of geopotential
height at 500 hPa (solid blue lines at 40 gpm intervals) and horizontal winds at 850 hPa
(wind barbs) at 2000 LST of 23 June 2018, which are both obtained from the ERA5
hourly reanalysis data. The purple rectangle indicates the location of the study area
shown in Figure 1b.

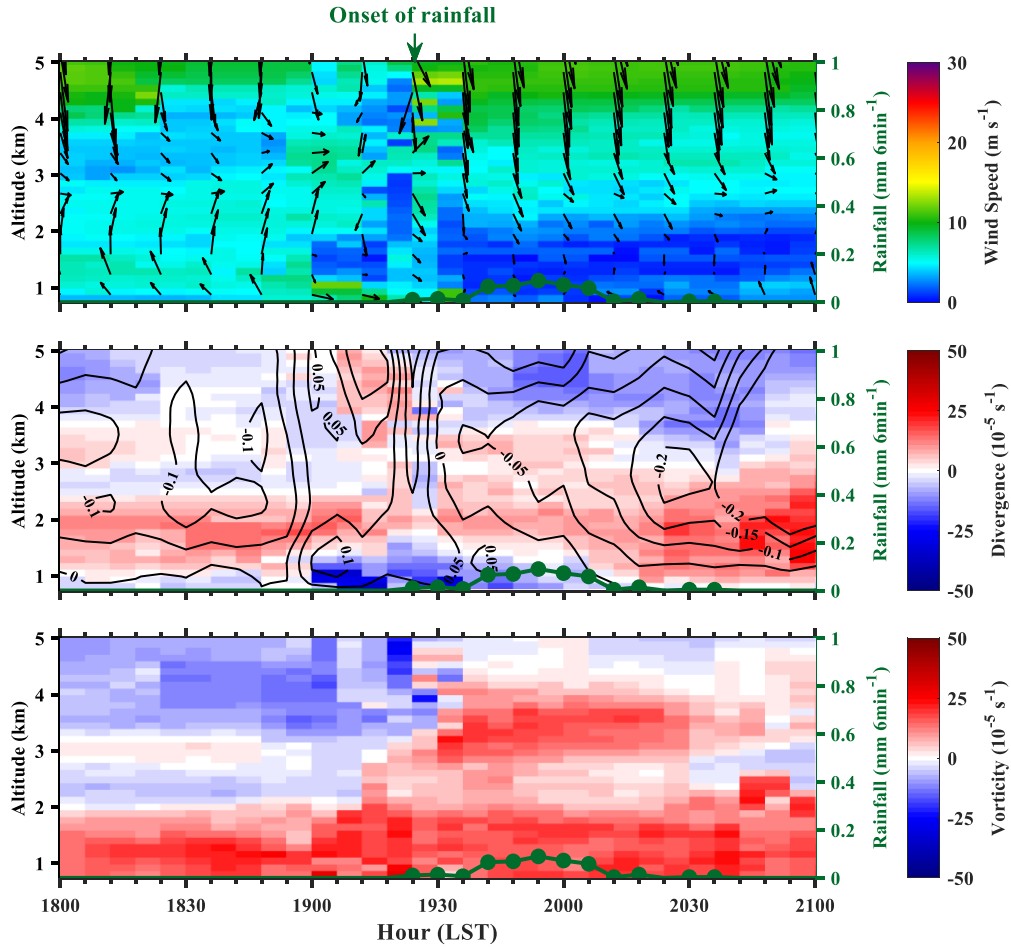

814

**Figure 9.** (a) Time series of horizontal wind vectors (m·s⁻¹) with wind speeds shaded

in the 0.5–5-km AMSL layer during the period of 1800–2100 LST 23 June 2018 at YQ

station. Green-dotted lines represent the triangle-area-averaged rainfall amount (mm

6min⁻¹) of triangle 1. (b) same as (a), but for the vertical profiles of the triangle-averaged

divergence (shaded, 10⁻⁵ s⁻¹) and vertical velocity (contour, m s⁻¹) for triangle 1. (c)

same as (b), but for the vertical profiles of the vorticity (shaded, 10⁻⁵ s⁻¹) for triangle 1.

821

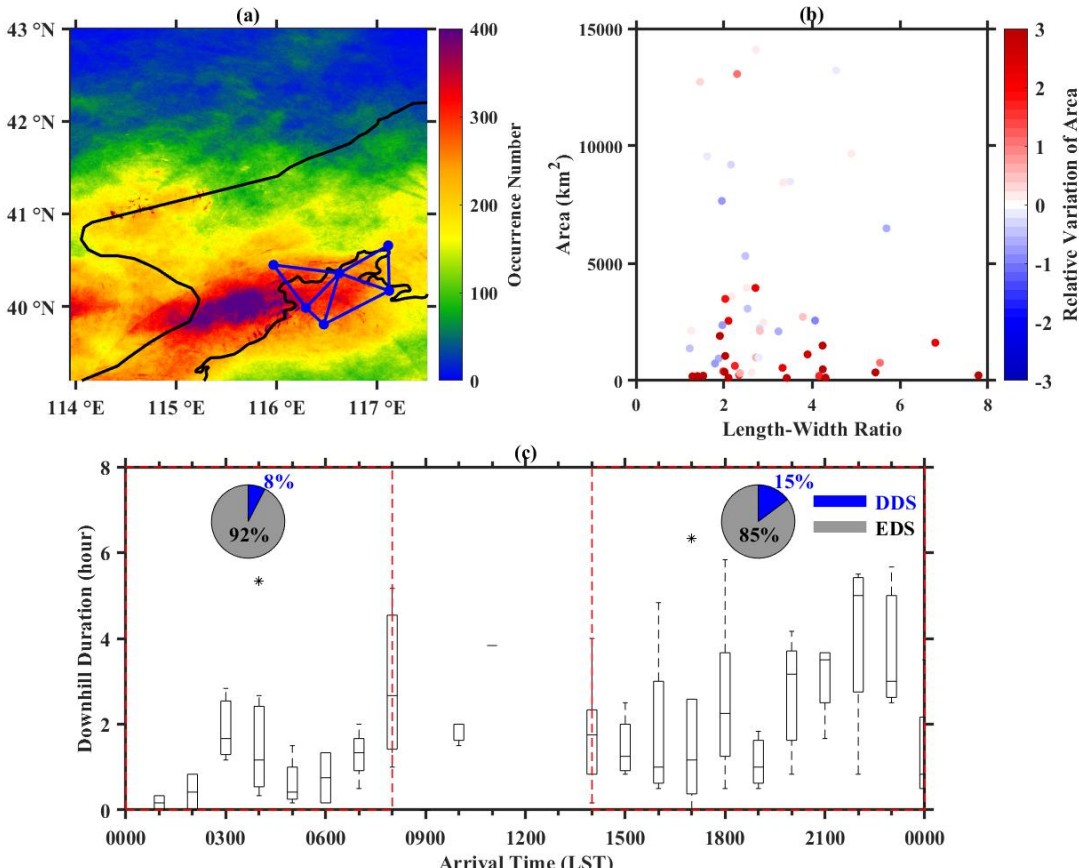

822

**Figure 10**. (a) The occurrence number (shaded) of reflectivity greater than 35 dBZ during downhill thunderstorm events. (b) Scatterplots showing the distribution of the initial length-width ratio and area of downhill thunderstorms, with the corresponding relative variation of area (shaded, km$^2$). (c) Boxplots showing the distribution of the arrival time and downhill duration of EDSs (red) and DDSs (blue). The central box represents the values from lower to upper quartile (25th–75th percentile), the vertical line extends from the 10th to 90th percentile, the solid line denotes the median.


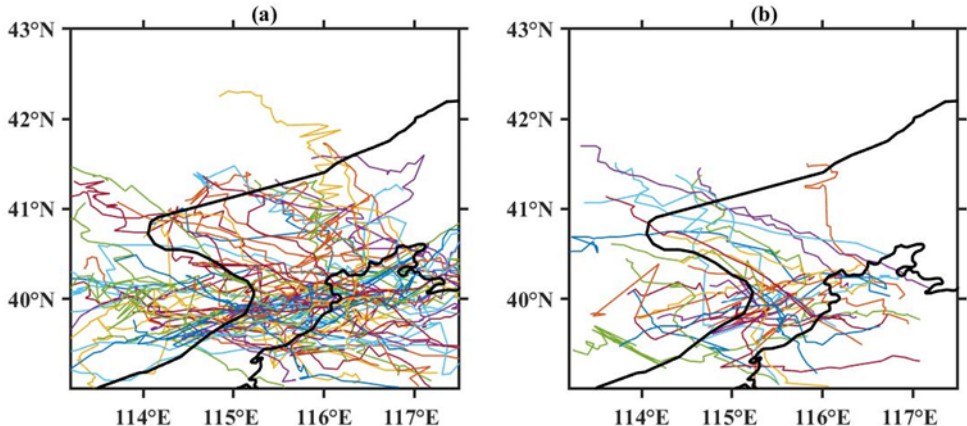


**Figure 11.** The trajectories (color shaded curves) of (a) 63 Enhanced Downhill Storms
(EDSs) and (b) 32 Dissipated Downhill Storms (DDSs). The bold black cure in the
middle marks the ridge line, and the bold black line in the lower right corner marks the
plain line that denotes the 200-m terrain elevation.

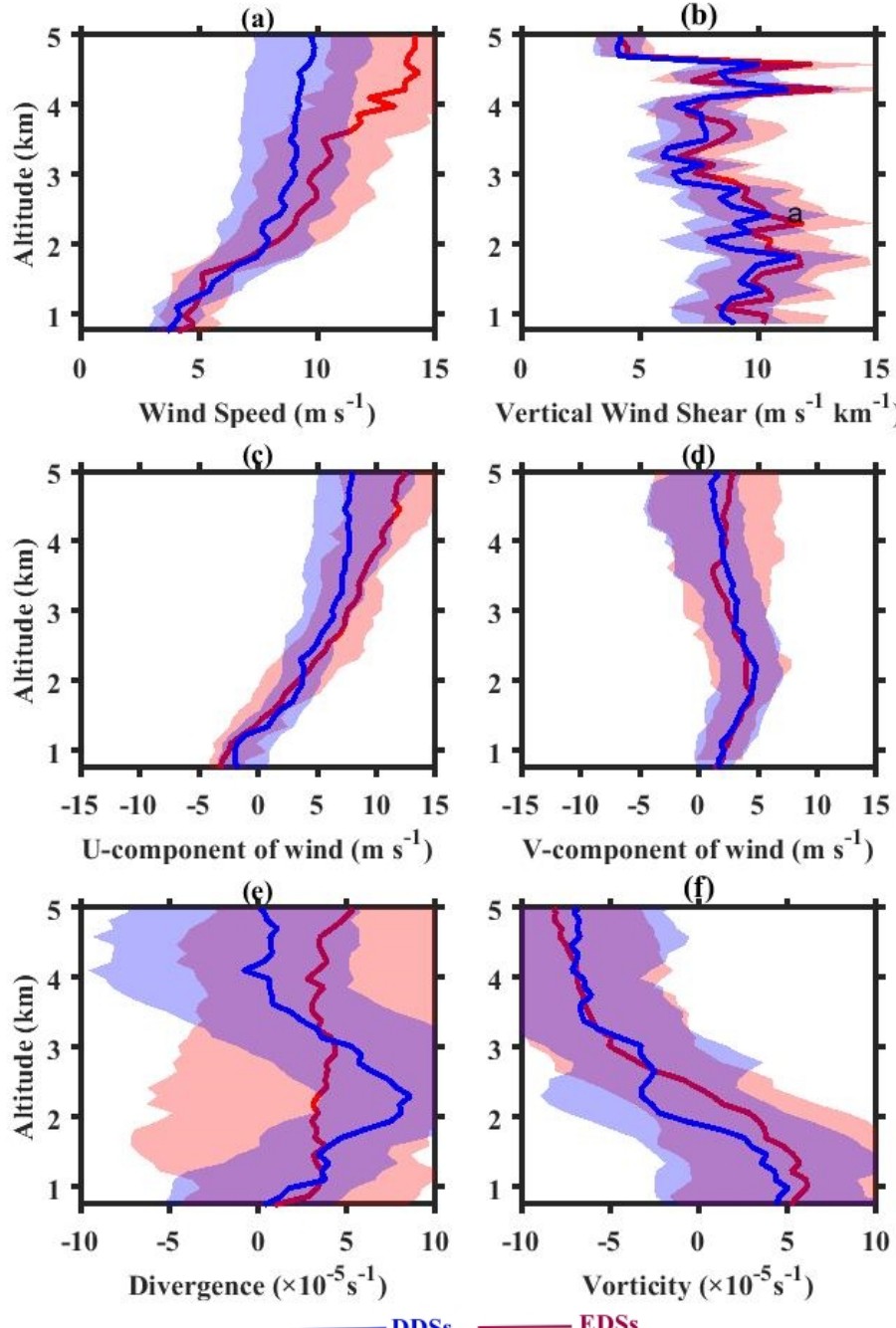


**Figure 12.** Vertical profiles of (a) wind speed, (b) vertical wind shear, (c) u-component
of wind, (d) v-component of wind over YQ station in two hours prior to the arrival of
EDSs (red) and DDSs (blue). (e) and (f) same as the above, except for the divergence
and vorticity over triangle 1 as shown in Figure 1b derived from the RWP mesonet,
respectively.