# Peer review of "Revisiting the evolution of downhill thunderstorms over"

_EGUsphere, 2024_

## Author Comment (AC1)

**Response to Reviewer #2's comments**

The authors undertook a comparative analysis of intensified and dissipating downhill thunderstorms utilizing a radar wind profiler mesonet. This research holds considerable importance for forecasting precipitation in Beijing. The comprehensive vertical observations facilitated by the radar wind profiler mesonet offer a valuable opportunity to scrutinize the thermodynamic and dynamic evolution mechanisms of downhill thunderstorms. While the manuscript is generally well-written, I have identified several concerns detailed below. Therefore, I recommend this paper would be accepted after major revision.

*Response: First of all, we appreciate tremendously your invaluable and constructive comments, which indeed help improving the quality of our manuscript. We have addressed the reviewers' concern one by one to the best of our abilities. For clarity purpose, here we have listed the reviewers' comments in plain font, followed by our response in bold italics, and the modifications to the manuscript are in italics.*

**Major comments:**

1.  The primary emphasis of this study lies in the examination of a case study on intensified downhill thunderstorms. However, the statistical analysis section appears somewhat abbreviated, considering its significance as the main focus of the research. To address this concern, the reviewer suggests expanding the discussion on the statistical analysis. In addition, it would be beneficial to select two representative cases each for both intensified and dissipated downhill thunderstorms.

*Response: Per your kind suggestion, the discussion on the statistical analysis has been expanded, including the trajectories and their moving directions of two types of downhill storms and the potential role of Beijing's topography and urbanization effects.*

*In addition, a case study for dissipated downhill thunderstorms has been added in this revision, please refer to the newly added section 4 entitled with "Comparison with a DDS event" for more details. We hope you are satisfied with these results.*

2. The proximity of Beijing's topography undoubtedly plays a pivotal role in the dynamics of downhill thunderstorms. However, the present study appears to provide less emphasis on the terrain effect. It is essential to delve deeper into how the terrain influences the dynamics throughout the processes under investigation.

*Response: Good point! The complex terrain of Beijing, comprising mountains and urban heat island (UHI), undoubtedly plays a pivotal role in the dynamics of downhill thunderstorms.*

*First of all, we elaborate on the possibility in the case study that the EDS and DDS events are influenced by small-scale variations of airflow in the narrow valley at the intersection of Mt. Taihang and Mt. Yan when they passed through $ROI_d$. In other words, the storms from northwest need to pass by the downslope, valley, and then upslope to reach the plain. The complex local terrain should be taken these factors into account for determining the evolution of thunderstorms during the southeastward propagation. However, the current resolution of observations is not capable of resolving the dynamic processes associated with the convective development in that region (Xiao et al., 2017). We hope to further explore this factor with the help of the numerical simulation in the future.*

*Secondly, the wind direction on the foothill is affected by both a mountain–valley-breeze circulation and urbanization effect (Dou et al.,2015). In the early morning, low-level westerly and northwesterly winds converged into the Beijing's plain area because of a combination of downslope mountain breezes and strong-UHI-induced convergence, which accelerate the speed of thunderstorms towards the plain. The weaker southeasterly upslope valley breezes in the late afternoon and evening make downhill storms slow down and contribute to the prolonged duration. This explanation also can be used to address your comment # 5, which is about the factors which contribute to the prolonged duration of downhill storms that reach plain areas in the late afternoon.*

*Finally, the urban-barrier effect is essential when the storms have entered the plain (Changnon, 1981; Zhang, 2020), especially under weak-UHI conditions. Thunderstorms may be bifurcated to move around cities due to the urban blocking*

***effects. The supply of moisture along south-southwesterly winds are also bypassed the urban center to produce minimum humidity. Thus, the storm may dissipate by an urban building-barrier induced divergence as it approaches the urban center, just like the EDS event.***

***To enhance the role of terrain, we have added a relevant paragraph in section 3.3 of this revision as follows:***

*"Interestingly, as the squall line propagated eastward and approached the urban center after 1500 LST, it rapidly dissipated with the area of convective echo decreasing by four fifths until 1600 LST (not shown). This appears to result from the blocking of water supply by the high risings over the Beijing's built-up area, the so-called "urban bifurcation" effects on moving thunderstorms (Changnon, 1981; Zhang, 2020). In this case, deep convection in the urban center and northern suburban area were suppressed due to the urban blocking effects. It was consistent with the persistent low-level divergence over triangle 3 and 4 with the maximum value of $3 \times 10^{-4}\ s^{-1}$ occurring near surface (not shown). Clearly, this result can help understand the urban building-barrier induced divergence and the dissipation of thunderstorm."*

***References:***

*Changnon, S.A.: METROMEX: a review and summary, Meteor.Monogr., No. 40, American Meteorological Society ,181, 1981.*

*Dou, J., Wang, Y., Bornstein, R., and Miao, S.: Observed spatial characteristics of Beijing urban climate impacts on summer thunderstorms. Journal of Applied Meteorology and Climatology, 54, 94–104, doi:10.1175/JAMC-D-13-0355.1., 2015.*

*Xiao, X, Sun, J., Chen, M. X., Qie, X., Wang, Y., and Ying, Z. M.: The characteristics of weakly forced mountain-to-plain precipitation systems based on radar observations and high-resolution reanalysis, Journal of Geophysical Research: Atmospheres, 122(6), 3193–3213, 2017.*

*Zhang, D.-L.: Rapid urbanization and more extreme rainfall events, Science Bulletin, 65, 516–518, https://doi.org/10.1016/j.scib.2020.02.002, 2020.*

3.  It would be beneficial for the authors to present the trajectories and their moving directions of two types of downhill storms.

*Response: Per your kind suggestion, we present the trajectories and their moving directions of two types of downhill storms in Figure 11. The western downhill area is found with high-frequency center mainly due possibly to the large amount of eastward propagation of most thunderstorms driven by the westerly or southwesterly flows during the warm seasons in Beijing. The difference of the trajectories and moving directions between EDSs and DDSs is unsignificant between two types of downhill storms.*

[Figure]

*Figure 11. The trajectories (color shaded curves) of (a) 63 Enhanced Downhill Storms (EDSs) and (b) 32 Dissipated Downhill Storms (DDSs). The bold black cure in the middle marks the ridge line, and the bold black line in the lower right corner marks the plain line that denotes the 200-m terrain elevation.*

*However, some downhill storms may arrive at the plain to the southwest of Beijing and not pass through the RWP mesonet. The dynamic parameters derived from the RWP mesonet are not enough to characterize the pre-convective environment for these downhill thunderstorms. Thus, we only select 68 downhill thunderstorms including 50 EDSs and 18 DDSs which passed through triangle 1 to the plain among all 95 samples and obtain the preceding divergence, vorticity and so on. To address your next question about original results of divergence and vorticity preceding EDSs and DDSs, the new results are described in the next question and revised correspondingly in the section 5.2.*

4. The distinctions in divergence and vorticity appear subtle, as indicated by the overlapping ranges of blue and red shadings between their maximum and minimum values in Figs. 8e and 8f. Specifically, the vorticity preceding Enhanced Downhill Storms (EDSs) appears weaker compared to that preceding Dissipated Downhill Storms (DDSs). The authors attribute this difference to the mountain-valley wind breeze. However, this explanation is challenging to grasp.

*Response: Per your critical comments, we try our best to further clarify this issue in this revision. The mean vertical wind profiles two hours prior to the arrival of the thunderstorms are investigated. Horizontal wind speed, vertical wind shear, u-component and v-component from the RWP in YQ, and divergence and vorticity over triangle 1 are calculated (Figure 12). Results indicate that wind speed preceding EDSs and DDSs is about 5 m s$^{-1}$ below 1.5 km (Figure 12a). Much stronger horizontal winds with the maximum wind speed exceeding 15 m s$^{-1}$ are observed in the 1.5-5 km layer in advance of the EDS events, The VWS below 5 km AMSL have no significant differences between EDSs and DDSs before their arrival (Figure 12b). But the VWS preceding EDS events is little bit stronger than that preceding DDS events, which could be likely associated the critical influence that high vertical wind shear exerts on convection. EDSs and DDSs mainly appears under the near-surface southeasterly and prevalent southwesterly low-level flow near the foothills. The persistent supply of water vapor is key for the successful propagation to the plains of downhill storms but doesn't determine the enhancement or dissipation of convection. Notably, the average v-component of wind decreases to near-zero above 3 km AMSL. The existence of stronger westerly flow above 3 km AMSL is a favorable condition for the intensification of downhill storms (Figure 12c), which well corroborates the results from case study.*

*The mean vertical structure of divergence and vorticity are given in Figure 12e and f. Before the arrival of downhill storms, one can see the presence of weak divergence near the surface due to the weak wind. Compared with EDSs, the divergence around 1.5-3 km AMSL is more evident near the arrival of DDSs with the*

*maximum value of $10^{-4}$ s$^{-1}$. When thunderstorms pass by, the strong divergence in the low level is not conducive to the extension of upward movement within the boundary layer which attributes to the dissipation of storms, especially when instability and moisture supply are unfavorable. In contrast, the high-level divergence at around 4-5 km altitudes promotes the compensation of the moist air and the upward transport heat, which ultimately reinforce the storm. The vorticity field in Figure 8f is characterized by cyclonic flows at lower-levels and anticyclonic flows at midlevel, which is possibly dependent on the synoptic forcing. The vorticity prior to EDSs seems to be stronger than that of DDSs, the cooperation between lower-level cyclones and less divergence of convective system tends to promote the maintenance of updrafts, leading to heavy rainfall.*

*The above-mentioned response and clarification have been well incorporated into this revised manuscript.*

5.   What factors contribute to the prolonged duration of downhill storms that reach plain areas in the late afternoon? Additionally, could you provide the number of cases that arrive at plain areas in the late afternoon and early morning, respectively?

*Response: To better observe the duration and arrival time about two types of downhill storms, we show the information at two hours interval with blue and red boxes for all 63 EDSs and 32 DDSs respectively (Figure 10c). Most of the DDSs arrive at the plain area in mornings and late afternoons. Specifically, 11 and 18 DDSs arrive at the plain area during the period of 0600-1200 and 1600-0000 LST which account for 34% and 56% of all DDSs, respectively. In contrast, EDSs tend to occur in early mornings and afternoons. 18 and 43 EDSs arrive at the plain area before 0800 LST and after 1400 LST, respectively, corresponding to the percentage of 26% and 68%.   Meso-scale circulations driven by the urban heat island (UHI) effect and topography may contribute to the difference of downhill storms' duration. As presented by Dou et al. (2015), the magnitude of UHI of Beijing at the nighttime are stronger than in daytime. In the early morning, low-level westerly and northwesterly winds converged into the*

*Beijing's plain area because of a combination of downslope mountain breezes and strong-UHI-induced convergence and accelerate the speed of thunderstorms towards the plain. The weaker southeasterly upslope valley breezes in the late afternoon and evening make downhill storms slow down and contribute to the prolonged duration. One caveat is that the conclusions may vary by the number of available sample cases.*

*The above-mentioned response and clarification have been well incorporated into this revised manuscript.*

[Figure]

*Figure 10. (a) The occurrence number (shaded) of reflectivity greater than 35 dBZ during downhill thunderstorm events. (b) Scatterplots showing the distribution of the initial length-width ratio and area of downhill thunderstorms, with the corresponding relative variation of area (shaded, km2). (c) Boxplots showing the distribution of the arrival time and downhill duration of EDSs (red) and DDSs (blue). The central box represents the values from lower to upper quartile (25th–75th percentile), the vertical line extends from the 10th to 90th percentile, the solid line denotes the median.*

6. The representativeness of this case needs clarification. While low-level convergence is highlighted as an effective signal for convective maintenance in this instance (Lines 33-34), this finding is not observed in the statistical section.

*Response: Thanks for your kind reminder. To clarify the representativeness of low-level convergence signal, we select a case for dissipated downhill thunderstorms with the similar trajectory of original case. The results show the effectivity of low-level convergence for convective maintenance. When the downhill thunderstorm approaches the plain, low-level divergence at the foot of the mountain is not conducive to the extension of upward movement within the boundary layer which may attribute to the dissipation of storms, especially when instability and moisture supply are unfavorable. This finding well corroborates the results in the statistical analysis in this revised manuscript.*

*Nevertheless, the above-mentioned dynamic features, which are necessary to diagnose the evolution of thunderstorms, are not adequate to fully characterize the environment in which downhill storms are embedded. In particular, more explicit analysis of thermodynamic parameters, such as CAPE, K index, precipitable water, will be performed to characterize the pre-storm environments in detail.*

*The above-mentioned response and clarification have been well incorporated into this revised manuscript.*

7. Lines254-256: A more comprehensive quantitative analysis is warranted to elucidate how cold-pool-induced horizontal vorticity overpowers over low-level wind shear before 1400 LST. Furthermore, could you provide an explanation for why this overpowering effect diminishes after 1400 LST?

*Response:Good point! We attempt to further examine the roles of cold pool and low-level wind shear in maintaining the intense squall line in accordance with the theory of Rotunno et al. (1988). However, given the difficulty in performing a comprehensive and quantitative analysis due to the inhomogeneous environment and measurement, here we qualitatively use the horizontal winds over YQ (Figure 6a) to estimate vertical*

**wind shear om the downslope and $T_{2m}$ to identify a cold pool (Figure 5). The added descriptions and discussion have been added to section 3.2, which is shown as follows:**

"*Further, we attempt to examine the roles of cold pool and low-level wind shear in maintaining the intense squall line in accordance with the theory of Rotunno et al. (1988). However, it's difficult to perform a comprehensive and quantitative analysis due to the inhomogeneous environment and measurement. Here, we qualitatively use the horizontal winds over YQ (Figure 6a) to estimate vertical wind shear (VWS) om the downslope and $T_{2m}$ to identify a cold pool (Figure 5). At 1300 LST, wind speed below 1.5 km AMSL was weaker than 5 m s$^{-1}$ while was stronger than 15 m s$^{-1}$ above 2.5 km AMSL. The maximum value of VWS occurred at the altitude of 1.8 km AMSL with the value exceeding 20 m s$^{-1}$ km$^{-1}$. In less than 10 minutes, cold downdrafts produced a sharp drop in $T_{2m}$ by 6°C in the south of the convective cells (Figure 5c-d). The effects of the resulting low-level VWS might balance with those of the cool pool, which helped stimulate the development of more intense storms from 1300 to 1330 LST. Meanwhile, the accompanying evaporative cooling in the descending flows strengthened the cold pool. After 1330 LST, horizontal wind speeds in the lowest 2 km layer strengthened to shrink the low-level VWS to about 10 m s$^{-1}$ km$^{-1}$. The cold-pool-induced horizontal vorticity could overpower that of the low-level wind shear, partly facilitating the dissipated radar echo before 1400 LST (Figure 5e). Moreover, this might be related to the relatively strong cold pool located in the south, which potentially cut off the warm southerly inflow from the plains to the mountains. Then, cool pool weakened with convection and the overpowering effect diminished.*"

**Minor comments:**

1.   The warm advection induced by veering of winds can also be observed in DDSs (Figs. 8c and 8d).

***Response: Yes, the warm advection induced by veering of a southwesterly wind at low level to a midlevel westerly wind is common in the warm season. It may be one of the factors that is beneficial to the downhill storms. But it is more controlled by the large-***

*scale synoptic circulation rather than meso-scale environment. Thus, we delete the discussion after careful consideration.*

2. The triangles 3-4 and the RWPs at BWO, PG, and SDZ are not utilized in the present analysis. Why then do authors reference the RWP mesonet?

*Response: Per your suggestions, we added related analyses based on the measurements from the triangles 3-4 in the case study of the EDS event, which is shown as follows:*

"*Interestingly, as the squall line propagated eastward and approached the urban center after 1500 LST, it rapidly dissipated with the area of convective echo decreasing by four fifths until 1600 LST (not shown). This appears to result from the blocking of water supply by the high risings over the Beijing's built-up area, the so-called "urban bifurcation" effects on moving thunderstorms (Changnon, 1981; Zhang, 2020). In this case, deep convection in the urban center and northern suburban area were suppressed due to the urban blocking effects. It was consistent with the persistent low-level divergence over triangle 3 and 4 with the maximum value of $3 \times 10^{-4} \, s^{-1}$ occurring near surface (not shown). Clearly, this result can help understand the urban building-barrier induced divergence and the dissipation of thunderstorm.*"

*Thus, the triangle 3-4 is utilized to provide the subsequent dynamic conditions after downhill storms enter the plain. In addition, the triangle 3-4 are available to monitor the downhill storms from the north or northeast even though there are relatively few samples of this type of thunderstorm.*

3. Line 332: "And" should be avoided at the beginning of a sentence in formal writing.
*Response: Removed as suggested.*

4. Lines 368, 370, and 376: The "figure 8b", "figure 8f", and "figure 8a" should be "Figure".
*Response: Amended as suggested.*

5. Line 31: which support -> supporting.

*Response: Amended as suggested.*

6. Figure 8: The legend of red and blue lines should be included in the figure.

*Response: The legends of red and blue lines have been added in the figure, per your kind suggestion.*

---

## Author Comment (AC2)

**Response to Reviewer #1's comments**

This paper provides an examination of the evolution of downhill thunderstorms over Beijing by using a radar wind profiler (RWP) mesonet. The results elucidate the storm's dynamic structures with a focus on both enhanced and dissipated events. The usage of high-resolution horizontal divergence and vertical motion data from the RWP mesonet to characterize the pre-storm environment represents a significant advancement in understanding these meteorological phenomena. The detailed case study, along with statistical analyses spanning the warm seasons of 2018 to 2021, contribute to our understanding of the factors influencing thunderstorm intensity and evolution in this region. Thus, I recommend the publication of this paper in Atmospheric Chemistry and Physics after some minor corrections for clarification.

*Response: We appreciated tremendously your positive and invaluable comments, which indeed help improving the quality of our manuscript. We have addressed the reviewers' concern one by one to the best of our abilities. For clarity purpose, here we have listed the reviewers' comments in plain font, followed by our response in bold italics, and the modifications to the manuscript are in italics.*

**Minor comments:**

1. The discussion on the implications of these findings for weather forecasting and model improvements may be further strengthened. Expanding this discussion could enhance the practical relevance of the research, suggesting pathways for incorporating these observations and methodologies into weather forecast models.

*Response: Thanks for your suggestion. In the previous work of our team, it has been confirmed that dynamical variables with higher temporal and spatial resolution derived from the RWP mesonet have great potential to improve the prediction skill of convection with the aid of a machine learning model. The results therein show that the usage of RWP observational data as the random forest model input tends to result in better performance in rainfall/non-rainfall forecast 30 min in advance of rainfall*

*onset than using the ERA-5 data as inputs. Per your kind suggestion, the reference has been cited and related discussion is further strengthened in section 5.2 as follows:*

*"In the previous work, it has been confirmed that these dynamical variables derived from the RWP mesonet in Beijing provide strong supports for machine-learning-based prediction of severe convection (Wu et al., 2023). The results therein show that the usage of RWP observational data as the random forest model input tends to result in better performance in the rainfall/non-rainfall forecast 30 min in advance of rainfall onset than using the ERA5 reanalysis data as inputs. In the future, these dynamic observations and methodologies need to be further incorporated into machine learning model for improving the prediction skill of downhill thunderstorms."*

*References:*

*Wu, Y., Guo, J., Chen, T., and Chen, A.: Forecasting Precipitation from Radar Wind Profiler Mesonet and Reanalysis Using the Random Forest Algorithm, Remote Sensing, 15, 1635. https://doi.org/10.3390/rs15061635, 2023.*

2.   As suggested by the statistical analysis, urbanization effects may play a potential role in the enhancement of downhill thunderstorms. The authors may include a more detailed discussion of the process-level mechanism due to the importance of this effect.

*Response: Per your suggestion, we add a more detailed discussion of urbanization effects on downhill thunderstorms as follows:*

**Wind direction on the foothill is affected by both a mountain–valley-breeze circulation and urban heat island (UHI) effect (Dou et al.,2015). In the early morning, low-level westerly and northwesterly winds converged into the Beijing's plain area because of a combination of downslope mountain breezes and strong-UHI-induced convergence, which accelerate the speed of thunderstorms towards the plain. The weaker southeasterly upslope valley breezes in the late afternoon and evening make downhill storms slow down and contribute to the prolonged duration. This explanation correlates well with the results in statistical analysis that the downhill storms that reach plain areas in the late afternoon have prolonged duration.**

***In addition, the urban-barrier effect is essential when the storms have entered the plain (Changnon, 1981; Zhang, 2020), especially under weak-UHI conditions. Thunderstorms may be bifurcated to move around cities due to the urban blocking effects. The supply of moisture along south-southwesterly winds are also bypassed the urban center to produce minimum humidity. Thus, the storm may be suppressed by an urban building-barrier induced divergence as it approaches the urban center, just like the EDS event. We have added a relevant paragraph in section 3.3 of the manuscript as follows:***

*"Interestingly, as the squall line propagated eastward and approached the urban center after 1500 LST, it rapidly dissipated with the area of convective echo decreasing by four fifths until 1600 LST (not shown). This appears to result from the blocking of water supply by the high risings over the Beijing's built-up area, the so-called "urban bifurcation" effects on moving thunderstorms (Changnon, 1981; Zhang, 2020). In this case, deep convection in the urban center and northern suburban area were suppressed due to the urban blocking effects. It was consistent with the persistent low-level divergence over triangle 3 and 4 with the maximum value of $3\times10^{-4}\ s^{-1}$ occurring near surface (not shown). Clearly, this result can help understand the urban building-barrier induced divergence and the dissipation of thunderstorm."*

*References:*

*Changnon, S.A.: METROMEX: a review and summary, Meteor. Monogr., No. 40, American Meteorological Society ,181, 1981.*

*Zhang, D.-L.: Rapid urbanization and more extreme rainfall events, Science Bulletin, 65, 516–518, https://doi.org/10.1016/j.scib.2020.02.002, 2020.*

*Dou, J., Wang, Y., Bornstein, R., and Miao, S.: Observed spatial characteristics of Beijing urban climate impacts on summer thunderstorms. Journal of Applied Meteorology and Climatology, 54, 94–104, doi:10.1175/JAMC-D-13-0355.1., 2015.*

3.	The methodology for identifying downhill thunderstorms has been clearly described with a thorough explanation of its criteria. It may be beneficial to clarify on how this methodology compares or improves upon existing approaches.

*Response: Thank you for your suggestion! How this methodology compares with or improves upon existing approaches has been added in section 2.1 as follows:*

*"Most of previous research, either case studies or small sample statistics analysis, lack an objective criterion used to determine downhill thunderstorms. They typically focus on EDS in the presence of high-impact weather and less consider DDS. Compared to the existing approaches in the literature, our methodology can discriminate between these two types of downhill thunderstorms for its capability in defining the timing and location of storms and tracking their corresponding evolution. Therefore, this methodology can be readily applied to other regions with similar topography as long as weather radar measurements are available."*

4.	The authors may strengthen the discussion regarding the pre-convective environment for the downhill thunderstorms. Some discussions for the statistical characterizations for the humidity and temperature profiles may be helpful.

*Response: The radiosonde launches just twice a day at 0800 and 2000 Local Standard Time (LST) at the station of BWO. For the sake of improving the prediction skill of summertime storm, an additional radiosonde launch is performed at 1400 LST daily at the BWO for the period from June 1 to August 31. In the case study of an EDS event, the thermal stratification is just obtained by the sounding 6–7 hours prior to storm. Thus, it's a pity that we cannot obtain the real-time humidity and temperature profiles from radiosondes for the pre-convective environment of the downhill thunderstorms. In further study, we will perform the statistics analysis of some thermodynamic parameters derived from radiometer, such as CAPE, K index, precipitable water, to characterize the pre-storm environments in detail.*